# Stem Cell-Based Disease Models for Inborn Errors of Immunity

**DOI:** 10.3390/cells11010108

**Published:** 2021-12-30

**Authors:** Aline Zbinden, Kirsten Canté-Barrett, Karin Pike-Overzet, Frank J. T. Staal

**Affiliations:** Department of Immunology, Leiden University Medical Center, 2333 ZA Leiden, The Netherlands; a.zbinden@lumc.nl (A.Z.); k.pike-overzet@lumc.nl (K.C.-B.); k.cante@lumc.nl (K.P.-O.)

**Keywords:** hematopoietic stem cells, induced pluripotent stem cells, immune deficiency, inborn errors of immunity, primary immune deficiency

## Abstract

The intrinsic capacity of human hematopoietic stem cells (hHSCs) to reconstitute myeloid and lymphoid lineages combined with their self-renewal capacity hold enormous promises for gene therapy as a viable treatment option for a number of immune-mediated diseases, most prominently for inborn errors of immunity (IEI). The current development of such therapies relies on disease models, both in vitro and in vivo, which allow the study of human pathophysiology in great detail. Here, we discuss the current challenges with regards to developmental origin, heterogeneity and the subsequent implications for disease modeling. We review models based on induced pluripotent stem cell technology and those relaying on use of adult hHSCs. We critically review the advantages and limitations of current models for IEI both in vitro and in vivo. We conclude that existing and future stem cell-based models are necessary tools for developing next generation therapies for IEI.

## 1. Introduction

Inborn errors of immunity (IEI, or primary immunodeficiencies [PIDs]) encompass a group of more than 400 inherited disorders that result in partial or complete loss of normal immune development or function [1]. They are almost all caused by monogenic mutations that affect the cells of the immune system. The clinical presentation is variable and includes severe infections, autoimmune diseases, and malignancies. Since these diseases tend to be rare, patient material is limited. Therefore, the study of these diseases requires either mouse models, which are useful but do not always phenocopy the human disease, or the use of human stem cell-based models in vitro or in vivo to recapitulate the disease. In general, two approaches are taken: (1) using patient-specific induced pluripotent stem cells (iPSCs) that carry the affected gene mutations, followed by further differentiation into the immune cell lineages concerned, or (2) use of adult hematopoietic stem cells (HSCs) that can be genetically manipulated and differentiated in vitro or in vivo after transplantation in immunodeficient mice. The term “adult stem cells” used throughout this review refers to stem cells found after birth, in opposition to embryonic stem cells (ESC), which are referring to stem cells found during development. In addition, it is important to note that ESC are pluripotent, having the capacity to develop into all three germ layers and cell types, while adult stem cells are specific to a certain organ or restricted cell types.

Stem cells have the ability to self-renew as well as differentiate along multiple cell lineages: features that can offer significant insights into developmental biology, disease modeling and regenerative medicine. Precisely, the reconstitution capacity of human HSCs (hHSCs) to efficiently repopulate both myeloid and lymphoid lineages have been of interest for decades to treat a number of immune diseases [2]. For instance, the transplantation of hHSCs is currently used for the care of hematologic malignancies such as leukemia, certain cellular immunodeficiencies and autoimmune diseases [3]. The rationale behind this approach is an initial broad lymphoablation of the defective immune cells eradiating the recipient’s immunological memory and subsequent infusion of functional hHSCs from donors, thereby allowing an extensive immunological renewal. One important drawback of allogenic hHSC transplantation, is the trigger of graft versus host disease (GvHD) with a cumulative incidence of acute GvHD of 40–60% in matched related or unrelated donors [4]. 

Over the last several decades, major technological advances, including in genome editing, have widened the possibilities to correct genetic defects that cause immune diseases. As a result, strategies have focused on using the patient’s own stem cells to manipulate them genetically by introducing genes or correcting genetic defects [5]. The use of autologous cellular material bypasses the current shortage issue of hHSCs collected from donors or patients and importantly, eliminates GvHD and the need for life-long immunosuppressive medication (Figure 1).

## 2. Paradigm of HSCs Origin and Potency: Implications for Disease Modeling 

The recapitulation of certain aspects of ontogenetic processes is the common denominator of most strategies that aim to differentiate pluripotent stem cells (PSCs) towards adult hematopoietic cells and subsequent lineage-committed cells. Therefore, drawing lessons from developmental biology is of utmost importance for the development of reliable disease models, both in vitro and in vivo. 

In the 50s, studies showed that mice that received a lethal dose of radiation could recover their blood cells with bone marrow (BM) cell infusion (reviewed thoroughly by Eaves et al. [6]). It was therefore established that BM had a long-term repopulation capacity. Since then, the HSC field has considerably progressed and it is now accepted that HSCs are defined by their in vivo function; that is, their self-renewal capacity and generation of all lineage-committed blood and immune subsets [7]. Interestingly, the scientific consensus slowly starts to shift from the early studies describing the HSCs as dormant at the top of the hematopoietic hierarchy regulated by a specific microenvironment in the BM, so-called HSC niche. There is now accumulating evidence that (1) HSCs are not the only producer of hematopoietic cells, (2) HSCs organigram is not a single hierarchy and may resemble more a holacratic network and (3) hematopoiesis isn’t restricted to BM and can be extramedullary (discussed below).

Embryonic hematopoiesis is driven by complex temporal and spatial patterns, which occurs sequentially as primitive hematopoiesis (so-called first wave) and definitive hematopoiesis (so-called second and third waves) [8]. Each wave is considered to have a distinct hematopoietic potential. Primitive hematopoiesis occurs in the embryonic yolk sac (YS) and primarily produces primitive erythrocytes, macrophages and megakaryocytes [9]. The second waves give rise first to multipotent progenitors, a process that is commonly termed erythro-myeloid progenitor (EMP) hematopoiesis. Subsequently, extraembryonic YS, aorta-gonad-mesonephros (AGM) region, and placenta generate multipotent progenitors with lymphoid potential, a subpopulation referred as lymphoid-primed multipotent progenitors (LMPP) [10,11]. EMP and LMPP are defined as HSC-independent. The origin of HSCs from a developmental point of view is, however, more controversial. In the late 90s, it was shown that HSCs first arise at the AGM region via endothelial-to-hematopoietic transition (EHT) [12,13], in contrast to the early postulate in the 70s, when HSCs were described to originate from the YS [14]. Later, evidence pointed out that HSCs from the AGM, major umbilical and vitelline vessel, as well as from the placenta emerged three days after the onset of the primitive hematopoiesis [15]. In addition, it was suggested that definitive HSCs emerged intra-embryonically within the para-aortic splanchnopleure, therefore disregarding the YS as source of HSCs [16,17]. After more than 50 years of debate around YS as source of HSCs, lineage tracing experiments using tamoxifen in mice could detect HSCs present in the YS at E7.5 that contributed to the HSC pool in adult mice [18,19]. Follow-up studies showed that the YS contributed up to 40% of the HSCs in fetal liver and adult BM [20]. These data are currently debated due to the long-term presence of tamoxifen in vivo [21,22]. The HSCs present at the AGM site are located in particular in the dorsal aorta (DA) [23], and due to their low cell numbers are considered transient without massive expansion [15,24,25]. At a later stage in mice, HSCs migrate and colonize the fetal liver where expansion can occur until they mobilize finally towards the BM and the thymus [26,27]. Of note, additional hematopoietic waves with HSC potential have been recently described, further adding complexity to the current model [28,29]. 

Although different sites have been identified as a source of HSCs during development, it remains unclear and controversial whether there is a common embryonic ancestor to all HSCs and from where exactly it would emerge. One common denominator of these hemogenic sites is the hemogenic endothelium (HE): a transient specialized endothelium with the capacity to initiate the process of EHT [30,31]. Lineage tracing experiments showed direct sprouting of HSCs from the ventral aortic endothelial, thereby reenforcing the postulate around their endothelial origin [32]. Hematopoietic cells arising from HE can be identified by the co-expression of vascular and hematopoietic markers CD34, VE-cadherin (C144), CD117, CD90 and CD105 in humans [33]. In mice, HSCs arising from HE have been defined by the expression of VE-cadherin, CD45, CD93, Kit, SCA1 and CD31 [34,35,36]. From early studies in mice, it was shown that HSCs originating from the AGM region are the earliest stem cells capable to engraft adult mice, while HSCs detected as early as E9 in the YS and para-aortic splanchno pleura had the capacity to engraft neonatal recipients but not adult mice [37,38]. These studies were the first to suggest the presence of immature HSCs (currently termed ‘pre-HSCs’) that require a specific fetal microenvironment for engraftment and further maturation. Some characteristics of these fetal microenvironmental cues could be mimicked in vitro and allow for the maturation of VE-cadherin^+^ E9.5-E.10 progenitor cells and subsequent engraftment in adult recipients [25,39,40,41]. Further research allowed additional maturation steps to be defined, termed, respectively, pro-HSC I and pro-HSC II. However, the question remains whether the hemogenic endothelial cells emerging at different embryonic sites are similar in terms of hemogenic potential (extensively reviewed by Lange et al. [42]). 

It is generally accepted that HSCs seed the BM around birth and, by following a hierarchical process, can produce HSC-derived cells. We also know that circulating HSCs and progenitor cells (HSPCs) can be recruited to particular sites where they participate in multiple ways to the local immune response. However, until recently HSCs were rarely found in a steady state outside of the BM. Tissue-resident phenotypic HSCs in a steady state have been found in humans in the periphery [43] in the spleen [44,45], liver [46,47] and intestine [48,49], and in animal models in the periphery [50,51,52,53], spleen [54], liver [55], lungs [56], gingiva tissue [57], skeletal muscle [58] and kidney [54]. However, the self-renewal capacity still needs to be examined, because it appears that these phenotypic HSC pools do not have the same reconstitution capacity. For instance, the intestinal HSCs have a bias towards lymphoid lineages as they co-expressed CD45RA, CD7 and CD56 but not the myeloid marker CD33 [48]. 

Elegant lineage tracing studies in mice further demonstrated the heterogeneity of HSC clones and clonal dynamics in vivo [59,60], which forced the scientific communities to revisit the previous postulate that long-term HSCs (LT-HSCs) are the major contributors to steady-state hematopoiesis in adults [61]. Sun et al. reported only 5% contribution from LT-HSCs to mature blood cells [59]. In a follow-up study, it was shown that megakaryocytes are the immediate progeny of LT-HSCs rather than arising from other lineages in the peripheral blood (PB) [62]. Brugman et al. used DNA barcoding to trace transplanted phenotypically defined HSCs in vivo [60]. They showed clonal restriction in the thymus, as only a small fraction (10%) of the barcoded HSCs were observed in the thymus. Laurenti et al. showed heterogeneity within HSCs pools based on differential expression of the cell cycle regulator CDK6 [63]. LT-HSCs lack CDK6 protein, delaying cell cycle progression and preserving the pool long-term. Another study identified in human BM a subpopulation with self-renewal characteristics similar to LT-HSC, but with a lymphoid bias and independent of HSC contribution [64]. 

## 3. HSC-Based Immune Disease Modelling

### 3.1. Source of Stem Cells: Important Considerations

Disease models aim to mimic molecular mechanisms representative of human in vivo physiology. The establishment of such models, both in vitro and in vivo, relies in part on the source of cells. To model IEI, one can utilize two main sources of HSCs: adult HSCs (primary cells) or PSCs that are induced to develop towards hematopoietic lineages. PSCs can be from embryonic origin (ESCs) or from reprogrammed somatic cells into iPSCs. Here we discuss important considerations in the use of adult stem cells and iPSCs. 

#### 3.1.1. Adult Hematopoeitic Stem Cells

Adult HSCs are primary cells isolated from patients and are a powerful tool to study disease etiology at the molecular level. They can be enriched from PB, BM or umbilical cord blood (UCB) using CD34^+^ as surface marker [65]. Although all these sources of cells contain CD34^+^ cells, they are evolving in different physiological niches. PB CD34^+^ cells are collected utilizing mobilizing agents such as granulocyte colony-stimulating factor (G-CSF) which may have repercussions on cell function, compared to the BM CD34^+^ cells isolated directly from the BM interstitial space [66]. In general, UCB has the highest percentage of CD34^+^ cells in the absence of mobilizing agents, but a drawback is the limited volume of cord blood after birth. Therefore, important differences are expected to be found within the composition of CD34^+^ cell subsets based on their isolation source (extensively review by Tajer et al. [67]). Overall, CD34^+^ cells from UCB retain a higher fraction of uncommitted progenitors compared to PB and BM [68]. BM CD34^+^ cells have more committed progenitors such as MLP/BLP compared to PB [69]. These findings are in agreement with clinical data that shows a lower number of UCB CD34^+^ cells is needed for engraftment; however, the complete immune reconstitution is slower compared to other sources of CD34^+^ cells due to the presence of uncommitted progenitors [70,71,72,73]. Similarly, it may explain why PB CD34^+^ cells engraft faster than BM CD34^+^ cells in a clinical setting [74]. 

A number of studies have also shown that T cells are present in higher numbers in CD34^+^ enriched products from PB compared to BM or UCB (reviewed by Panch et al. [75]). In addition, the proportion of regulatory T cells (Tregs) was lower in PB CD34^+^ cells compared to BM CD34^+^ cells [76]. This may explain the result of a meta-analysis comparing BM or PB CD34^+^ cells for allogenic HSC transplantation in 1521 adults with acute leukemia: transplanted PB CD34^+^ cells were associated with a higher rate of GvHD [77]. Mature donor T cells transplanted within the PB subsets can be activated by the post-transplantation microenvironment and trigger significant damage to the host tissues [78]. In contrast, Tregs play an important role to maintain immunological self-tolerance and immune homeostasis [79]; and when infused together with conventional T cells in patients who received an allogenic CD34^+^ graft, Tregs successfully suppressed GvHD [80]. Of note, a recent study by the Kupper group is adding complexity to the GvHD interplay; they have identified that tissue-resident memory T cells from the host survive the pre-conditioning and contribute to the GvHD in allogenic HSC transplantation [81]. 

The heterogeneity of HSCs has a direct impact for disease modelling and translational medicine in terms of engraftment and reconstitution capacity. Moreover, the heterogeneity of HSCs can also affect transduction efficiency. Genovese et al. showed how different CD34^+^ subsets isolated from PB had differential lentiviral transduction efficiencies, with primitive progenitor subset (CD34^+^ CD133^+^ CD90^+^) having the lower transduction efficiency compared to early progenitors (CD34^+^ CD133^+^ CD90^−^) and committed progenitors (CD34^+^ CD133^−^) [82]. Recently, Radtke et al. performed a side-by-side comparison of most commonly enriched HSC subsets: CD34^+^ CD133^+^, CD34^+^ CD38^low/−^ and CD34^+^ CD90^+^ [83]. They used phenotypic, transcriptional and functional readouts to characterize each subset and identified the CD34^+^CD90^+^ subset as having the least committed progenitors. The CD34^+^CD90^+^ subset was also the smallest sub-population of about 5% of CD34^+^ cells. This highly enriched population had the potential for short and long-term multilineage engraftment [83,84]. In this study, the CD34^+^CD90^+^ were successfully transduced with a high efficiency (in contrast to the previous study [82]), indicating this subset as a reasonable candidate for gene therapy and transplantation applications. 

In vivo studies further showed a bias towards certain lineages from CD34^+^ HSC clones, including myeloid, lymphoid, megakaryocytic lineages [85,86], while others showed a balanced reconstitution [87,88]. Similar results with myeloid-biased HSCs and lymphoid-biased HSCs (mostly T cells) were found in human clinical settings [89]. The authors tracked CD34^+^ cells (isolation from BM and PB) over time in patients who received gene therapies for Wiskott-Aldrich syndrome (WAS), sickle cell disease (βS/βS) and thalassemia (β0/βE). The study also revealed a highly polyclonal nature with about 50–200 000 HSCs contributing to mature blood cells. Although the cohort study was small, longitudinal tracking did provide considerable information regarding the hierarchies of human hematopoiesis. The polyclonal nature of HSC clones in human was also demonstrated in a recent study by tracking mitochondrial mutations within CD34^+^ cells collected from patients [90]. The contribution of HSCs from BM to mature blood cells was detected in B, T, NK and myeloid cells. Interestingly, with aging it was shown that BM HSCs become biased toward myeloid lineages in humans [91]. 

The complexity of lineage bias is highlighted by a recent study from the Weissman group, where HSC clones were tracked using barcodes in mice [92]. They showed important differences in lineage commitment between different pre-transplantation conditioning regimens: irradiation and anti-ckit antibody treatment (i.e., ACK2) of Rag2^−/−^
*IL-2Rγ*^−/−^ mice. Three distinct groups were observed in conditioned animals representing balanced, myeloid- and lymphoid-biased subsets, while no lineage bias was observed when conditioning was absent. The underlying mechanism behind the various biases remains unclear and the characterization of each of these subsets challenging. 

These studies highlight the need for a robust identification, characterization and enrichment of HSC clones with defined and optimal properties. In addition, there is a need to define a better consensus around the markers used to define the different hematopoietic populations, which currently varies greatly among studies. It will subsequently reduce the variability seen in pre-clinical and clinical studies. 

#### 3.1.2. Differentiation Potential and De Novo Generation of HSCs from iPSCs

De novo generation of HSCs is highly interesting for regenerative medicine as it would provide an unlimited source of cells for clinical applications. Indeed, long-term expansion of ex vivo isolated human HSCs is still challenging, in contrast to murine expansion that has now been well established [93]. This is mostly the results of important mechanistic differences between mouse and human development and adult hematopoiesis. In addition, generating HSCs from autologous iPSCs has addresses the issue of GvHD. However, the current unknowns around the HSC origin, heterogeneity and plasticity makes the establishment of a defined differentiation protocol for PSCs challenging (i.e., iPSC, ESCs and adult CD34^+^ stem cells). Two main strategies are possible to differentiate stem cells along the hematopoietic lineage: (1) directed differentiation that sequentially uses cell-extrinsic factors in medium or from co-cultures, and (2) direct conversion and forward programming that use defined transcription factors introduced intrinsically. In some case, both strategies can be combined. Whereas the embryonic hematopoietic waves are temporally and spatially separated in vivo, the in vitro complete recapitulation of these complex patterns, so that they coexist, has proven difficult. Early differentiation protocols aiming to generate HSCs were more resembling the first transient HSC-independent wave of embryonic hematopoiesis rather than definitive hematopoiesis. One additional challenge is the lack of specific markers that could discriminate all the specific cell types from the different hematopoietic waves [94,95]. 

The use of more defined culture conditions has enabled significant progress towards more reliable production of hematopoietic cells. Analysis of signaling pathways have pushed our knowledge forward. For instance, recent studies have shown that the modulation between activation and repression of signaling pathways can directly impact on the production of hematopoietic progenitors resembling either primitive extra-embryonic progenitors or definitive-like hematopoietic progenitors [96,97,98]. Primitive hematopoietic cells (defined as CD235a^+^) are temporally dependent on the activin-nodal pathway, while the production of definitive-like progenitors (expressing *RUNX1*) is not [98]. Activation of the Wnt pathway while blocking the TGFβ pathway triggers the production of definitive-like progenitors marked also by upregulation of *HOXA* genes [97]. In addition, the extensively debated use of the HE as an intermediate step for iPSC-derived hematopoietic cultures has considerably improved the differentiation outcomes [99]. However, the scientific community agrees that these PSC-derived cells via directed differentiation, so far, are generating HPCs rather than HSCs. These PSC-derived HSC-like cells have failed to show multilineage engraftment capacity and multilineage differentiation abilities in vitro [100]. Overexpression of transcription factors is one approach to improve the HSC differentiation; forced expression of *FOSB*, *GFI1*, *RUNX1* and *SPI1* in endothelial cells could reconstitute primary and secondary mouse recipients [101]. Other transcription factors with differentiation potential were identified by Doulatov et al. as *ERG*, *HOXA9*, *RORA*, *SOX4* and *MYB*, which were used in CD34^+^CD45^+^ myeloid precursors to generate multilineage progenitors in vivo [102]. However, a bias towards myeloid lineages was observed and only short-term engraftment was achieved. Interestingly, identification of the epigenetic regulator *EZH1* by Vo et al. may improve the current differentiation protocols [103]. They showed how *EZH1* is an active epigenetic repressor of hematopoietic multipotency in early mammalian embryo, therefore being an attractive target to enhance differentiation. Moreover, the knockdown of *EZH1* unlocked lymphoid potential in differentiating ESCs in vitro. Sugimura et al. performed a large screen for transcription factors and identified *ERG*, *HOXA5*, *HOXA5*, *HOA10*, *LCOR* and *RUNX1* as sufficient to generate HSC-like cells from isolated HE. These derived-HSC-like cells could engraft in primary and secondary recipients and lead to the reconstitution of myeloid, B- and T-cell lineages [104]. Using single-cell sequencing, Fidanza et al. compared iPSC-derived HSC-like cells with their in vivo counterparts and identified small populations of cells that were clearly different between the two groups [105]. They hypothesized that these differences in gene expression are responsible for the functional differences observed between in in vitro HSC-like cells and their in vivo counterparts. One additional consideration regarding the use of iPSCs is the iPSC epigenome. For instance, Cypris et al. compared the epigenetic profile of iPSCs differentiated towards hematopoietic lineages with adult CD34^+^ stem cells isolated from UCB, PB and BM [106]. Principal component analysis of DNA methylation profiles clearly showed separated clustering of adult CD34^+^ stem cells and iPSC-derived HSC-like cells. 

Detailed reviews of differentiation steps and variations among protocols to generate HSCs and blood cells from PSCs (ESCs and iPSCs) have been reviewed elsewhere [8,42,100,105,107]. Overall, generated cells from iPSCs can be differentiated further towards a number of hematopoietic lineages, however, with a much lower yield compared to the HSCs obtained from isolated CD34^+^ cells from patients. Importantly, terminally differentiated hematopoietic cells from PSCs exhibit important differences and behave differently compared to their counterpart in vivo; and compared to their primary-derived or embryonic counterparts (discussed below). Finally, clinical translation ideally requires the development of protocols using transgene-free cells, which remains currently out of reach. 

### 3.2. In Vitro Inborn Errors of Immunity (IEI) Disease Models

The manipulation of HSCs combined with technological advances in gene editing is an attractive strategy to study human immune cell development and generate relevant IEI disease models. In vitro IEI disease models have, so far, focused on differentiation towards T cells, NK cells, granulocytes and monocytes (Figure 2). Severe combined immunodeficiency (SCID) is characterized in patients by a lack of T cells, accompanied in some cases by a B and NK-cell defect (SCID pathophysiology reviewed by Kumrah et al. [108]). Briefly, a number of genetic aberrations cause SCID: mutations in the interleukin-2 receptor common γ-chain (*IL-2Rγ*) or *JAK3* lead to a T^−^B^+^NK^−^ phenotype in patients (i.e., lack of T and NK cells, but presence of B cells). Mutations in the adenosine deaminase (ADA) gene leads to a T^−^B^−^NK^−^ phenotype, while mutations in the RAG1 or RAG2 genes lead to a broader range of phenotypes.

Studies aiming to model SCID in vitro have focused mainly on using stem cells (adult CD34+ stem cells or iPSCs) that are differentiating towards lymphoid lineages [109,110,111,112,113]. Chang et al. reprogrammed primary keratinocytes obtained from SCID patients with a JAK3 mutation [109]. They differentiated iPSCs towards T cells using the OP9-DLL4 co-culture system and identified an early block in the development of the JAK3-SCID iPSCs. After 14 days, only small populations of CD7^+^CD16^−^CD56^−^ T cells and CD7^+^CD16^+^CD56^+^ NK cells were present in the JAK3-iPSCs compared to iPSCs from a healthy donor, and *JAK3*-mutant cells did not differentiate further into CD4^+^/CD8^+^ double-positive (DP) or single-positive (SP) T cells. The differentiation arrest at or before the DN2 stage correlates to in vivo studies [114] and could be lifted by restoring the mutation using CRISPR/Cas9. Similarly, Menon et al. looked at SCID-X1 iPSCs carrying a mutation in the IL-2Rγ, as well as genetically corrected iPSCs using TALEN along the T and NK-cell differentiation paths using the OP9-DLL1 system [110]. They could successfully restore the T and NK-cell differentiation of patient-derived cells to a level similar to the control iPSC line. Here, they showed the presence of CD4^+^CD8^+^ DP but no data regarding the presence of TCRαβ was provided. In contrast to the study from Chang et al., Menon et al. included ESCs as an additional positive control. Interestingly, the differentiation efficiencies towards CD16^+^CD56^+^ NK cells dropped in iPSCs compared to ESCs, highlighting the differences between the two types of stem cells. 

The impact of null RAG2 mutation could be assed in vitro by using RAG2-SCID iPSCs on an OP9-DLL1 system [111]. The strongest block in T-cell development was observed from early CD7^−^CD5^−^ to CD4^+^CD8^+^ cells. NK-cell development was also affected and an increase in CD7^−^CD56^+^CD33^+^ cells could be observed. Almost no CD4^+^CD8^+^ DP could be observed in the RAG2-SCID iPSCs (~2%) and no TCRδ rearrangement could be observed for CD7^+^CD56^−^ sorted cells. The low numbers of CD4^+^CD8^+^ DP in RAG2-iPSCs were an aberrancy because they did not have TCR rearrangements, consistent with the absence of RAG1-RAG2 activity. Themeli et al. also generate a repaired line, which had restored capacity to induce T-cell differentiation and TCR rearrangement. NK-cell development could also be rescued [111]. RAG2 loss of function was also investigated in the 3-dimensional artificial thymic organoid (ATO) culture system [113]. ATO are co-cultures of murine stromal cells expressing DLL1 or 4 with CD34^+^ cells and better mimic the 3-dimensional microenvironment [115,116]. Indeed, the T-cell development on the OP9-DLL1 and -DLL4 2-dimensional stromal layer is quite poor and at best leads to varying percentages (10–70%) of DP cells and virtually no SP cells [109,110,111]. Using an ATO system, Gardner et al. obtained higher efficiencies with ~90% CD4^+^CD8^+^ DP and subsequently ~60% CD3^+^ TCRαβ^+^ cells [113]. RAG2-iPSCs generated in the ATO system had a similar block in T-cell development between CD7^−^CD5^−^ DN and CD4^+^CD8^+^ DP; however, with a higher percentage of remaining RAG2 CD4^+^CD8^+^ DP with no TCR rearrangement (~40%). These residual CD4^+^CD8^+^ DP cells could also be observed in iPSCs carrying various RAG1 mutations (up to ~10%) [112]. In this study, Brauer et al. used an OP9-DLL4 system with three RAG1 patient-specific iPSC lines: two were associated with a SCID clinical phenotype and one with the Omenn syndrome. In both SCID and OS cells, the block was described at an early stage in T-cell development, but with significant development of DP cells. This is in contrast to other reports in which CD34^+^ cells from patients were used, in such cases recombinase deficiency results in a complete block at the CD5^+^CD7^+^ DN stage without any CD4^+^CD8^+^ DP cells developing [117]. We previously have shown that hypomorphic RAG1 CD34^+^ patient cells transplanted into humanized mice in the context of preclinical gene therapy development for RAG1-SCID generated some CD4^+^CD8^+^ DP cells without CD3 [118], demonstrating that hypomorphic mutations and full null mutants in recombinase activity result in different developmental arrests, i.e., a full block at the CD5^+^CD7^+^ DN stage and a later arrest at CD3^−^DP stage in an hypomorphic situation. Overall, it remains unclear whether these RAG1/2 CD4^+^CD8^+^ DP may be an artefact of in vitro iPSC based system in which T-cell development follows unusual pathways or potentially pathways that resemble fetal lymphopoiesis or if the RAG1/2 mutations would be hypomorphic and not resulting in a total loss of function of the recombinases.

In vitro models focusing on lymphoid lineage differentiation were also used to look at specific defects in T and NK-cell development due to mutation in the WAS gene [119]. WAS encode for a protein (known as WASp) that is known to activate actin polymerization required to generate actin filaments. WASp is specific to hematopoietic lineages and plays an important role in the intercellular synaptic communications. WASp deficiency is referred to as Wiskott-Aldrich syndrome and is characterized by a defect in T cells and NK cells [120]. Laskowski et al. generated iPSCs from WAS patients and differentiated them using the OP9-DLL1 co-culture system [119]. WAS-iPSCs, genetically corrected WAS-iPSCs and control ESCs progressed from CD34^+^CD43^−^ endothelial cells to CD34^+^CD45^+^ HSC-like cells with the same efficiencies, indicating that the transition via the HE is not altered by the WAS mutant. CD56^+^CD7^+^ and CD56^+^CD94^+^ NK cells were significantly reduced in WAS-iPSCs compared to ESCs and corrected iPSC line. Interestingly, the few WAS-mutant NK cells showed a functional defect with upregulated interferon-γ (IFNγ) and tumor necrosis factor α (TNFα) secretion. WAS-mutant iPSC showed a defect in the generation of CD4^+^CD8^+^ DP and CD3^+^TCRαβ cells. Of note, the expression of CD34^+^ during T-cell differentiation from the control iPSCs remained high, as opposed to T-cell differentiation from CD34^+^ cells isolated from patients. The persistence of CD34 expression is indicating a more immature phenotype for iPSC-based T-cell differentiation. In addition, none of the studies differentiating iPSCs have included primary adult CD34^+^ stem cells as positive control for their differentiation. 

IEI disease modelling with a focus on innate cells has been possible. For instance, chronic granulomatous disease (CGD) is caused by mutations in the gene coding for NADPH oxidases that are involved in the respiratory burst in phagocytic leukocytes [121]. As a result, CGD patients are suffering from recurrent bacterial and fungal infections due to a defective phagocytic function. Several studies have looked at modeling CGD in vitro using CGD-iPSCs [122,123,124,125,126,127,128]. Utilizing different genetic corrective strategies, such as zinc-finger targeted insertion [122,129], TALEN [124,127], BAC transgenesis [126], or CRISPR/Cas9 [125,128], it was possible to correct CYBB and NOX2 genes encoding for one of the NADPH oxidase subunits. NADPH oxidase function could be successfully restored and using bacterial killing assays, they demonstrated the rescued phagocytic function of the iPSC-derived macrophage or granulocyte. Differentiation from hPSCs towards macrophages relies on a standard protocol using IL-3 and M-CSF [130]. Modification and refinement using Stem Cell Factor (SCF), VEGF or BMP-4 has further optimized the differentiation of various subset of macrophages [131,132,133,134]. Lachmann et al. showed how either granulocytes or macrophages could be generated from iPSCs by either using M-CSF (for macrophages) or G-CSF (for granulocytes) [134]. However, the question remains whether the functionality in vitro and in vivo of iPSC-derived macrophages resemble adult macrophages or rather embryonic macrophages from the primitive hematopoietic waves. Data from Takata et al. and Buchriser et al. are pointing out that iPSC-derived macrophages may resemble macrophages arising from the YS-hematopoiesis [132,133]. 

### 3.3. In Vivo Disease Models: Towards a Fully Hummanized Immune System

In vitro differentiation capacity towards hematopoietic lineages from adult CD34^+^ stem cells, ESCs or iPSCs remains limited compared to their in vivo counterparts. For example, the in vitro T-cell development protocol can generate CD4^+^CD8^+^ DP (usually in low numbers) that have been positively selected for their TCR; however, no studies until now have recapitulated negative selection in vitro, a process essential for immune tolerance. In addition, rarely are B-cell development and other alternative lineages looked at in IEI in vitro models, despite the development of in vitro protocol [136]. In vivo immunodeficient mouse models and subsequent humanized mouse models offer the possibility to study the human immune system to the extent that, so far, isn’t possible in vitro.

There is a large number of immunodeficient mouse models that have been developed since the establishment of the nude mice by Flanagan in the 60s [137,138]. Nude mice phenotype relies on the disruption in the *Foxn1* gene resulting in the lack of a thymus and subsequent impairment in the development of the adaptive immune system. Although nude mice have been extensively used as a model for tumor growth, they are failing to reconstitute the human immune system due to the presence of B cells and innate cells, such as highly active NK cells. As an alternative, Scid mice carry a mutation in the *Prkdc* gene—a protein necessary for non-homologous end joining of ds-DNA—, resulting in impaired B and T-cell development [139]. Scid mouse models are also considered to have B and T-cell leakiness (i.e., spontaneous generation of T and B cells), to be very sensitive to radiation, and have low engraftment of transplanted human PBMC and HSCs [140,141]. In contrast, the Rag1 or Rag2 deficient mice do not show this immunoglobulin leakiness or sensitivity to radiation [142,143]. Deletion of the Rag1/2 genes cause an arrest in the rearrangement of the T-and B-cell receptors and therefore leads to a lack of T and B differentiation [142,143]. Rag1^−/−^ and Rag2^−/−^ mice have been used as pre-clinical models for Rag1 and Rag2 gene therapies, as they exhibit similar phenotypes to patients suffering from SCID [118,144,145]. In a study from Garcia-Perez et al., BM CD34^+^ cells from Rag1^−/−^ mice were isolated and genetically corrected via lentiviral transduction of codon-optimized *Rag1* gene under an MND promoter [118]. Rag1^−/−^ mice that received the transplantation of Rag1-transduced CD34^+^ cells had a reconstitution of the T- and B-cell development. This preclinical study and other previous work were the drivers of the NCT04797260 clinical trial for the treatment of RAG1-SCID, which is currently recruiting patients in Europe for corrective RAG1 gene therapy [146]. 

Although Rag1^−/−^ mice haven been of great use to establish gene RAG1 therapy, they do not cover the high heterogeneity of the disease. Indeed, Rag1 deficiency in human is caused by mutations in the Rag1 genes and can be classified in 4 groups: (1) mutations that destabilize the tertiary structures such as for Zinc binding, (2) mutations important for DNA binding, (3) catalytic RNase H-like domain and (4) mutations involved in the RAG1/RAG2 interaction [147]. This heterogeneity in mutations results in a broad spectrum of clinical and immunological phenotypes [148]. Tackling this issue, Ott de Bruin et al. generated a number of RAG1 mouse models using CRISPR/Cas9 targeting specific regions of the *RAG1* genes [149]. For instance, they targeted the region of RAG1 around residues R838-N852, which is a region with unknown functions. Sever phenotypes in mice with in frame deletion from 832–877 were observed with a block in B- cell and T-cell development at the pre-pro/pro-B cells and DN3, respectively. Interestingly, in silico modeling and algorithms led to contradictory predictions for the specific mutation H836Q. In vivo modelling of the H836Q mutation showed intact B- and T-cell development, while in silico modeling predicted deleterious effects on Rag1 function. These data highlight the need for functional in vivo assays to confirm or refute in silico models. 

The Rag1^−/−^ and Rag2^−/−^ mouse models can also retain a high level of NK-cell activity and have been shown to have an overall limited engraftment of human HSCs compared to more recent models [142,143,150]. For instance, one important breakthrough in the field occurred with the development of the IL-2Rγ^−/−^ mouse models [151]. The interleukin-2 receptor common γ-chain is part of a dimer forming receptors for several cytokine interleukins and is present on developing immune cells [152]. The targeted knockout of IL-2Rγ^−/−^ impairs the development of NK cells in addition to that of T and B cells [153]. In humans, X-linked SCID (SCID-X1) is caused by a mutation in the IL-2Rγ and results in presence of very low amount of T and NK cells, as well as lack of response to the common γ-chain cytokines, while B cells are not affected. Due to these similarities, the IL-2Rγ^−/−^ mouse background has been used to assess in vivo safety for corrective gene therapies for IL-2Rγ mutations [154,155]. For example, Poletti et al. used a IL-2Rγ^−/−^ Rag2^−/−^ mouse model as preclinical model for a gene therapy for X-linked SCID (SCID-X1) patients [154]. Of note, combining IL-2Rγ^−/−^ mutation with a Rag1^−/−^ or Rag2^−/−^ background is advantageous, as IL-2Rγ^−/−^ single knockout mice have been described to be leaky.

Another major breakthrough was the development of the non-obese diabetic NOD.scid.IL-2Rγ^−/−^ (NSG) mouse models [156,157]. While NOD mice have a strong predisposition to develop spontaneous autoimmune diabetes, the NOD-Scid mice do not develop diabetes, but have a decreased NK-cell activity and a higher engraftment of human HSCs. The combination of these features with the IL-2Rγ^−/−^ mice makes the NSG mice a powerful model to look at the human immune system in vivo with applications for personalized medicine and translation of immune-based therapies. NSG mice do not show any leakiness towards B and T cells [158], and enable long-term studies as they are surviving beyond 15 months without developing thymic lymphomas [159]. In several studies, immune reconstitution following the human developmental stages from transplanted HSCs could be observed in the NSG mice for both T and B cells, as well as myeloid and plasmacytoid dendritic cells [156,157,160,161,162]. Importantly, NSG mice humanized with CD34^+^ cells from patients carrying mutations are a powerful alternative to look at specific developmental defects, as mouse models with the corresponding mutations do not always recapitulate the extent of the disease phenotype. For instance, the IL-7R^−/−^ mouse model has B and T-cell deficiencies [163], whereas mutations in the IL-7 receptors in human leads to a T-cell deficiencies without B- or NK-cell developmental arrest [164]. NSG mice were used as a model to look specifically at T-cell development arrest caused by the transplantation of CD34^+^ cells obtained from SCID patients [117]. The authors showed that T-cell arrests caused by IL-7 receptor α, as wells IL-2rγ, were occurring in immature thymocytes much earlier than expected compared to studies with corresponding genetic mouse models. In 2018, Tan et al. used NSG mice to look at engraftment capacity of iPSC-HSC-like cells and compared the outcome to adult CD34^+^ stem cells [165]. They used the transient expression of the fusion transcription factor MLL-AF4 during the differentiation of iPSCs towards hematopoietic lineages to improved differentiation efficiencies. The transplantation of from iPSC-HSC-like cells into NSG mice reestablished myelopoiesis and lymphopoiesis without lineage bias. Although both adult CD34^+^ stem cells and iPSC-HSC-like cells achieved long-term engraftment, only the iPSC-HSC-like cells were prone to leukemic transformation. Whether this is due to the use of MLL-AF4, which is an oncogenic fusion protein in acute myeloid leukemia and B-cell acute lymphoblastic leukemia or due to the nature of iPSCs remains unclear. Additional arguments that have been proposed are that there is a high oncogenic risk from genetic and epigenetic aberration that a due to reprogramming; and that NSG mice provide a permissive environment, which combined would allow transformation of iPSC-derived cells [166,167].

From the NSG mouse background, a number of new mouse models were developed with improved engraftments and lineage reconstitution properties. For instance, the NSG-SGM3 have a triple knock-in for the human stem cell factor (SCF), granulocyte-macrophage colony-stimulating factor (GM-CSF) and interleukin-3 (IL-3), and the MISTRG mouse model that have the knock-in for macrophage colony-stimulating factor (M-CSF), GM-CSF, IL-3, thrombopoietin and signal regulatory protein alpha (SIRPα) [168,169,170]. However, the NSG, NSG-SGM3 and MISTRG mice still require pre-conditioning before robust humanization with CD34^+^ cells. Different inflammatory pathways are involved during pre-conditioning that can damage the BM microenvironment, the extent of which is not yet fully understood [171]. The successful humanization with CD34^+^ cell without pre-conditioning was demonstrated in the NOD,B6.SCID IL-2rγ^−/−^ kit^W41/W41^ (NBSGW) mice by Hess et al. [68]. NSGW41 mice are developed from the NSG by addition the kit^W41/W41^ allele, which results in the competitive advantages for the human transplanted cells [172]. Hess and colleagues also showed how the source of CD34^+^ cells (BM, PB or UCB) have a direct impact on the immune reconstitution in NBSGW mice, which is an important experimental consideration. In a follow-up study, Adigbli et al. demonstrated that the NBSGW mice could develop LT-HSC population from engrafted human UCB CD133^+^ cells without pre-conditioning [173]. In 2021, Coppin et al. generated mice expressing human IL-7 on a NSGW41 background mice, termed NSGW41hIL7 [174]. The expression of human IL-7 in these humanized mice leads to improved human T-cell development compared to the NSGW41 mice. 

Although the NSG mice and NSG-derived models are considerably improved models for studying the humanized system, one must have the awareness that humanization can have unintended immunological consequences, as described by two recent studies from Blümich et al. [175] and Janke and colleagues [176]. From the examination of a large cohort of humanized and non-humanized NSG mice with CD34^+^ cells, Blümich et al. raised some important considerations, such as the poor correlation between engraftment level in the humanized mice and the presence of human circulating CD45^+^ cells in the PB [175]. Subsequently, one can ask weather measuring engraftment based on PB is a reliable measure for such model. The study also showed that background lesions present in the non-humanized mice were exacerbated by the humanization with CD34^+^ cells, which indicate some level of GvHD [177]. Janke et al. looked at large cohort of NSG-SGM3 mice humanized or non-humanized with CD34^+^ cells [176]. They observed 3 main features (1) mast cell hyperplasia, (2) eosinophil hyperplasia and (3) hemophagocytic lymphohistiocytosis/macrophage activation syndrome (HLH/MAS)–like disease. Importantly, naïve and humanized NSG-SGM3 mice exhibited important differences in these three features.

Therefore, a number of alternatives to immunodeficient mouse models, including rats, pigs, dogs, non-human primates or zebrafish [178,179,180]. Overall, larger animals have a longer lifespan than smaller animal models. Longer lifespans may be resourceful for disease modeling and assessing therapies and treatment options under pre-clinical settings, however it has a direct effect on research in term of costs and research speed (i.e., breeding time). Rats can be up to 10 times larger than mice, thereby can receive transplants of larger sizes. As in humans, rats express CD4 and/or CD8 on macrophages and MHC class II on endothelial cells, which is not the case in mice, mimicking therefore closer human physiology. Interestingly, no study has reported a shorter lifespan for immunodeficient rats, in contrast to immunodeficient mice such as NOD-derived models, which have a high incidence of spontaneous thymic lymphoma [181]. A number of immunodeficient rat models exists with mutations in *Foxn1*, *Rag1*, *Rag2*, *Prkdc*, *Il2rg* or *IgM*; and could be humanized via the transplantation of PBMCs or enriched CD34^+^ cells [182,183,184] (review extensively by Adigbli et al. [178]). Pigs have also been used to generate immunodeficient lines carrying either spontaneous or induced mutations in *Rag1*, *Rag2*, *Artemis* or *Il2rg* (review extensively by Boettcher et al. [179]). As showed by Dawson et al. [185], pigs have fewer unique immunological genes compared to human or mice, and could better replicate and predict clinical outcomes in humans, which in turn have the potential to improve the clinical translation [186]. While immunodeficient pigs can accept a number of xenografts, immune humanization in pigs remains a constant challenge and further work is required [187]. The use of canine models for HSC transplantation have been recently reviewer by Graves et al. [180]. 

## 4. Future Perspectives

The development of reliable disease models is challenging, as it requires mimicking the human pathophysiology in vitro or in vivo (Table 1). For decades, in vivo models have provided considerable insights into physiology and disease mechanisms of a large number of IEI. Important breakthroughs related to the humanization of the immune system in mouse models have addressed some of the species-specific limitations of classic mouse models, bridging the gap between animals and humans. There is no denying that data and insights provided by in vivo models have been and are essential to the development of novel therapies and cures for human diseases; however, as Justice et al. simply explained: “a model is simply that: it is not the human” [188]. Researchers must have deep knowledge and understanding about the large variety of in vivo models in order to adequately answer specific scientific questions. In parallel, enormous efforts are invested in the field of in vitro models to move from classic monolayer cultures to 3-dimensional human co-culture systems that aim to provide platforms to study human-specific developmental processes and disease mechanisms in greater details. This includes the technical advances in perfused microphysiological systems, so called organ-on-a-chip technology (OoC), stem cell technology and gene therapy. Although in vitro models for IEI are still in their infancy, OoC and 3-dimensional culture systems are gaining considerable momentum in other fields and may significantly impact the immunology field in the coming years. For instance, some progress has been made with the use of artificial thymic organoids (ATO) to study T-cell development. Bosticardo et al. used ATO to confirm that severe T-cell lymphopenia from patients with DiGeorge syndrome was caused by extrinsic stromal defects (i.e., thymic stromal cells) [189]. Such in vitro screening methods are highly valuable to determine the nature of the T-cell deficiency and to direct treatment options at an early onset. ATO could be used to address the developmental blocks in T-cell development caused by mutations in genes such as *RAG1* and *RAG2*, ultimately giving important insight into the nature of the deficiency (i.e., hypomorphic situation or null mutations). Further optimization of the ATO system is required to switch from mouse stromal cell lines (such as MS5-DLL1/4) used to mimic the thymus stromal parts to a human source of cells that would better recapitulate the interactions between thymic epithelial cells and developing thymocytes. Indeed, in the study from Bosticardo et al., the correct in vitro development of TCRαβ^+^CD3^+^ from CD34^+^ cells isolated from a patient with ADA deficiency confirmed the extrinsic, metabolic origin of the deficiency in patients, highlighting that MS5-DLL4 cells in the ATO restore ADA activity and support T-cell development. Thus, the ATO only reveals intrinsic development defects. Efforts are currently invested in deriving thymic epithelial cells from hPSCs that can support human T-cell development in vitro and in vivo [190,191,192,193]. This is not only highly beneficial to improve current IEI models with genetic defects present on the hematopoietic counterpart, but also for immune deficiencies with extra-hematopoietic defects such as DiGeorge syndrome. 

To conclude, the success of disease modelling relies on the optimal combination of independent models, both in vitro and in vivo, as well as a critical interpretation of results and careful comparison to data of patients included in clinical studies. 

## Figures and Tables

**Figure 1 cells-11-00108-f001:**
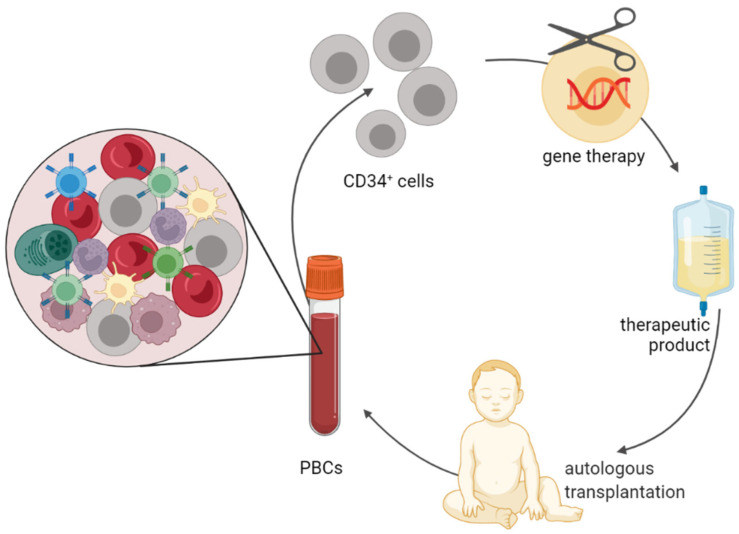
Overview of autologous HSC transplantation and corrective strategies for inborn errors of immunity using gene therapy. Created with BioRender.com.

**Figure 2 cells-11-00108-f002:**
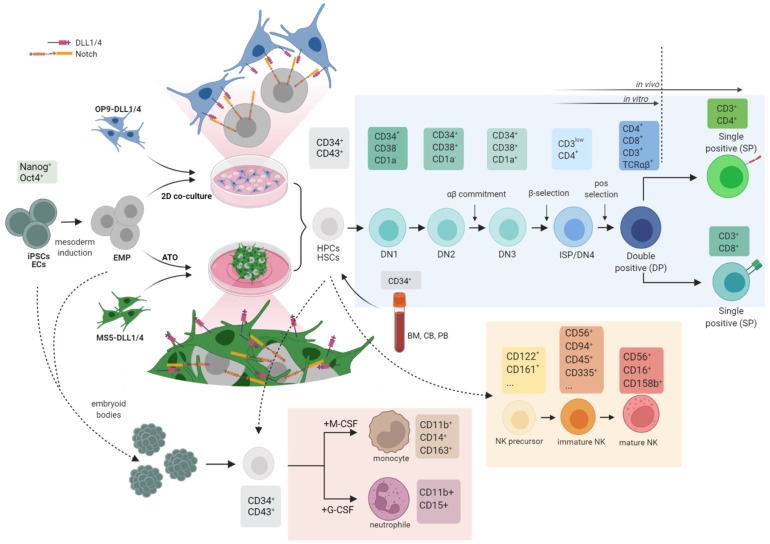
Overview of the current strategies used in IEI in vitro models to generate lineage-committed hematopoietic cells. So far, IEI in vitro models have focused on T-cell, NK-cell, monocyte and granulocyte (i.e., neutrophil) differentiations. T-cell differentiation can be performed in 2-dimension via the co-culture of mouse stromal cells (OP9) expressing delta-like ligands (DLL1 or DLL4) to stimulate Notch signaling on CD34^+^ cells. T cells can be differentiated in a 3-dimensional system via the use of artificial thymus organoids (ATO), where mouse stroma cells (MS5) expressing DLL1 or DLL4 are mixed with CD34^+^ cells. Generating NK cells in vitro is controversial; NK cells can be obtained via the OP9-DLL1/4 or ATO system but verification of their development via surface markers is currently challenging [135]. Generation of innate cells such as monocytes and neutrophils is usually performed via the formation of embryoid bodies which are induced towards mesoderm and subsequent hematopoietic progenitor cells; M-CSF and G-CSF are allowing the conversion towards one or the other lineage. CD34^+^ cells can be obtained from BM, UCB, PB as an adult stem cell source or via the differentiation of iPSCs or ESCs. BM: bone marrow, UCB: umbilical cord blood, PB: peripheral blood, EC: endothelial cell, EMP: erythroid-myeloid progenitor. Created with BioRender.com.

**Table 1 cells-11-00108-t001:** Summary of key considerations regarding models for IEI.

	Adult Stem Cells	hESCs	hiPSCs
Stem cell type	- Potential lineage bias - No ethical concern - GvHD for allogenic transplantation PBSC - Higher proportion of T cells in CD34^+^ enriched produced compared to BM or UCB - Lowest proportion of Treg - Transplantation outcomes with highest probabilities of GvHD BM - Most committed source of cells with high percentage of CMP/CLP - Faster engraftment UCB - Highest number of CD34^+^ cells compared to BM or PB - Highest number of uncommitted progenitors - Limited volume/quantity - Slower engraftment	- Lineage bias - Ethical concern - Low availably - GvHD	- Lineage bias - No ethical concern - Can bypass GvHD - Epigenomes considerations
		- Differentiation protocol are usually not xenofree (feeder cells or matrices) - Directed differentiation leads to a less mature phenotype compared to direct conversion and forwards programming - Direct conversion and forward programming involves the use of transgenes - Poorer engraftment properties compared to adult stem cells	
	In vitro IEI models	In vivo IEI models	
Model Architecture	- Potential for in-depth molecular and mechanistic characterization - Potential to generate fully human-based models - No ethical concerns - Low costs - Parallelization and high-throughput potentials - Focus mostly T-cell development - Reduced differentiation capacity in vitro compared to in vivo counterparts	- Systemic effects - Study of the differentiation and reconstitution capacity of all lineages possible - Ethical concerns - Time consuming and expensive - Fundamental biological differences with humans	

## Data Availability

Not applicable.

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
