# Peer review of "Stem Cell-Based Disease Models for Inborn Errors of Immunity"

_cells, 2021, doi:10.3390/cells11010108_

Round 1
Reviewer 1 Report
This manuscript concerns a review on disease models for inborn immune-mediated diseases. The authors presented an interesting review on in vivo and in vitro models. It takes into account many literature reports with a total of 182 cited references, including 36 references published in the last 3 years. The article is well written and designed in clear and concise manner. I appreciate the story on the HSC origin and potency, with the main known developmental process and controversies. Each time, the authors have endeavored to present not only the advantages of each model but also the remaining challenges as well as the debatable points, providing an important light in order to properly appreciate each models for IEI.
My opinion is that this review would be useful and would be accepted for publication in Cells
Author Response
Answer: We thank the reviewer for the many positive remarks.
Reviewer 2 Report
This is the much talked about research in regenerative medicine concerning haematopoietic stem cells. This article discusses the developmental origins, heterogeneity and subsequent implications for disease modelling, as well as the pros and cons of models based on induced pluripotent stem cell technology versus models relying on adult hHSCs, and reviews the advantages and limitations of current in vivo and in vitro models of IEI, however this article still has some shortcomings
1 The field of stem cells, a hot and frontier area of medical research, has developed rapidly, especially in recent years, yet the references in this article are somewhat outdated. A total of 182 references are cited in this article, of which only 38 are from the last three years
2 The article emphasises the different haematopoietic origins, but does not address the extent to which the haematopoietic origins have influenced the construction of disease models, the haematopoietic components of the latest disease models and where there is room for improvement and, specifically, where to overcome.
3 The article also analyses the heterogeneity of haematopoietic stem cells, but as above, it does not specifically address the extent to which the heterogeneity of mouse or adult haematopoietic stem cells affects the construction of disease models, and areas to be overcome in the disease model.
4 The article compares adult stem cells and induced pluripotent stem cells, but the content is somewhat disorganised and a detailed breakdown using tables is recommended. It is also recommended that the differences in differentiation between PSC and iPSC-derived haematopoietic stem cells be compared using a table.
5 The article describes existing in vitro IEI disease models, but does not summarise them effectively. It is suggested that a table incorporating the above-mentioned differences in haematopoietic stem cell origin, heterogeneity and differentiation be used to detail the origins, strengths, weaknesses and potential breakthrough points of existing models of haematopoietic stem cell-derived disease, based on an adequate summary.
Author Response
1 The field of stem cells, a hot and frontier area of medical research, has developed rapidly, especially in recent years, yet the references in this article are somewhat outdated. A total of 182 references are cited in this article, of which only 38 are from the last three years
Answer: We appreciate the reviewers’ suggestions, but what we have tried to do is give a concise but comprehensive overview of the field by acknowledging the primary literature in the field which now has a considerable history of over 15-20 years. Hence, we have cited much of the primary literature. Citing the original work and studies is also to avoid that the readers must jump from one review to another to find the original studies. We have now 43 references from the last three years.
2 The article emphasises the different haematopoietic origins, but does not address the extent to which the haematopoietic origins have influenced the construction of disease models, the haematopoietic components of the latest disease models and where there is room for improvement and, specifically, where to overcome.
3 The article also analyses the heterogeneity of haematopoietic stem cells, but as above, it does not specifically address the extent to which the heterogeneity of mouse or adult haematopoietic stem cells affects the construction of disease models, and areas to be overcome in the disease model.
Answer 2-3: We have now more thoroughly discussed the differences that may arise from using adult HSCs, with differences depending on the source of HSCs used (BM, UCB, PBSC) and the use of iPSC-derived HSCs that resemble fetal HSCs and therefore may lead to other lineage differentiation patterns and quantitative outcomes compared to adult HSCs. We have summarized this in Table 1.
4 The article compares adult stem cells and induced pluripotent stem cells, but the content is somewhat disorganised and a detailed breakdown using tables is recommended. It is also recommended that the differences in differentiation between PSC and iPSC-derived haematopoietic stem cells be compared using a table.
Answer: Detailed reviews on differentiation protocols from PSCs have been already reviewed elsewhere, we have included these references to the text lines 260-261. Reviewing in detail the differences in differentiation protocols would have made the review much longer and was not the focus of this review.
5 The article describes existing in vitro IEI disease models, but does not summarise them effectively. It is suggested that a table incorporating the above-mentioned differences in haematopoietic stem cell origin, heterogeneity and differentiation be used to detail the origins, strengths, weaknesses and potential breakthrough points of existing models of haematopoietic stem cell-derived disease, based on an adequate summary.
Answer: We thank the reviewer for this comment and suggestion, we have added Table 1 summarizing the key considerations regarding IEI models, including the comparisons between different stem cell sources (line 535).
Reviewer 3 Report
This is a well written review of stem cell models for IEI.
Comments:
Consider including in the abstract what is actually reviewed
Review the use of "adult" before hematopoietic stem cells. Some of the applications are in children, and the word 'adult' might not be used when HSC are obtained from cord blood.
In contrast to mice, human X-SCID does allow B cell.
Consider including canine models (SCID, LAD)
Author Response
Answer: We thank reviewer 3 for his comments and suggestions. We have specified clearly the definition of “adult stem cells” and reviewed the manuscript for a correct use of the term (line 35-39). A clarification regarding B cell development in human and mice X-SCID has been added (line 417-419). Finally, a new section regarding alternatives to mouse models and key references has been included line 482-500..